# Consequences of adaptation of TAL effectors on host susceptibility to *Xanthomonas*

**Doron Teper**, **Nian Wang***

Citrus Research and Education Center, Department of Microbiology and Cell Science, Institute of Food and Agricultural Sciences, University of Florida, Lake Alfred, Florida, United States of America

* nianwang@ufl.edu

**Data Availability Statement:** All relevant data are within the manuscript and its Supporting Information files.

**Funding:** NW received funding from the US Department of Agriculture-National Institute of Food and Agriculture (USDA-NIFA) Plant Biotic

## Abstract

Transcription activator-like effectors (TALEs) are virulence factors of *Xanthomonas* that induce the expression of host susceptibility (S) genes by specifically binding to effector binding elements (EBEs) in their promoter regions. The DNA binding specificity of TALEs is dictated by their tandem repeat regions, which are highly variable between different TALEs. Mutation of the EBEs of S genes is being utilized as a key strategy to generate resistant crops against TALE-dependent pathogens. However, TALE adaptations through rearrangement of their repeat regions is a potential obstacle for successful implementation of this strategy. We investigated the consequences of TALE adaptations in the citrus pathogen *Xanthomonas citri* subsp. *citri* (*Xcc*), in which PthA4 is the TALE required for pathogenicity, whereas *CsLOB1* is the corresponding susceptibility gene, on host resistance. Seven TALEs, containing two-to-nine mismatching-repeats to the EBE$_{PthA4}$ that were unable to induce *CsLOB1* expression, were introduced into *Xcc pthA4*:Tn5 and adaptation was simulated by repeated inoculations into and isolations from sweet orange for a duration of 30 cycles. While initially all strains failed to promote disease, symptoms started to appear between 9–28 passages in four TALEs, which originally harbored two-to-five mismatches. Sequence analysis of adapted TALEs identified deletions and mutations within the TALE repeat regions which enhanced putative affinity to the *CsLOB1* promoter. Sequence analyses suggest that TALEs adaptations result from recombinations between repeats of the TALEs. Reintroduction of these adapted TALEs into *Xcc pthA4*:Tn5 restored the ability to induce the expression of *CsLOB1*, promote disease symptoms and colonize host plants. TALEs harboring seven-to-nine mismatches were unable to adapt to overcome the incompatible interaction. Our study experimentally documented TALE adaptations to incompatible EBE and provided strategic guidance for generation of disease resistant crops against TALE-dependent pathogens.

## Author summary

Mutation of the EBEs of susceptibility (S) genes via genome editing and utilization of naturally occurring EBE variants have been used to generate disease resistant plants.

Interactions Program under grant no. 2017-67013-26527 (https://urldefense.proofpoint.com/v2/url?u=https-3A__nifa.usda.gov_&d=DwIGaQ&c=sJ6xIWYx-zLMB3EPkvcnVg&r=t-amc4JbEo_7rK5LJaQISQ&m=UwUad2YpFlH0cZEyGMwyy_77saJljw-DsIzVlOnbpUE&s=vTNJqcH7NJaoJpwXqJtD5o4VhgdZvnfA_9mh1VFSCtA&e=). DT received funding from BARD, the United States - Israel Binational Agricultural Research and Development Fund, Vaadia-BARD Postdoctoral Fellowship Award No. FI-562-2017 (https://urldefense.proofpoint.com/v2/url?u=https-3A__www.bard-2Disus.com_&d=DwIGaQ&c=sJ6xIWYx-zLMB3EPkvcnVg&r=t-amc4JbEo_7rK5LJaQISQ&m=UwUad2YpFlH0cZEyGMwyy_77saJljw-DsIzVlOnbpUE&s=9CNBr-wDo6GKbQSgmIQWM7Tg7Ww-lvtCCLnQhio3gNKU&e=). The funders had no role in study design, data collection and analysis, decision to publish, or preparation of the manuscript.

**Competing interests:** The authors have declared that no competing interests exist.

However, TALE adaptations may lead to resistance loss, limiting the long-term efficacy of the strategy.

We utilized an experimental evolution approach to test TALEs adaptations in the *Xanthomonas citri*-citrus pathosystem using designer TALEs that cannot recognize the EBE of host targets. We identified adaptive TALE mutations and deletions that occurred during less than 30 cycles of repeated infections, which reconstituted the virulence on the host. Adaptive variants originated from TALEs that harbored a small number of mismatches (≤5) to the EBE, whereas designer TALEs that harbored larger number of mismatches (≥7) to the EBE failed to adapt in the duration of this study. Our study experimentally demonstrates adaptive rearrangements of TALEs during host adaptation and suggests that the potential durability in the resistance of modified crops should be a significant factor to be considered prior to their introduction into the field.

## Introduction

Transcription activator-like effectors (TALEs) are bacteria-encoded eukaryotic transcriptional activators delivered into host cells through the type III secretion system (T3SS) [1]. TALE protein architecture contains N-terminal T3SS secretion and translocation signal, central DNA binding domain and C-terminal eukaryotic acidic transcriptional activation domain and nuclear localization signals (NLS) [1]. The DNA binding domain is composed of an array of 1.5–33.5 nearly identical tandem repeats of 33–34 AA [1, 2]. The 12th and 13th amino acids of each repeat, known as the "repeat-variable diresidue" (RVD), vary between repeats and dictate the affinity of each repeat to an individual nucleotide [3]. Through this recognition mechanism, the TALE repeat array determines the binding specificity of each TALE to a DNA sequence located in the promoter of host target genes that serves as an effector-binding element (EBE) [4].

*Xanthomonas* is one of the most economically important plant pathogens infecting most plant species [5]. TALEs are key virulence factors in numerous *Xanthomonas* spp. [6]. *Xanthomonas* TALEs induce the expression of host susceptibility (S) genes to cause disease [7]. The number of TALEs in different *Xanthomonas* bacteria varies from 0 (the majority of pepper and tomato infecting strains) to close to 30 (as found in *X. oryzae* pv. *oryzicola*) [7, 8]. While most non-TALE effectors of *Xanthomonas* are usually associated with disruption and manipulation of host defense signaling [9–11], TALEs were reported to target more diverse cellular functions. For example, multiple *X. oryzae* pv. *oryzae* (*Xoo*) TALEs induce the expression of rice *SWEET* sugar transporter genes to facilitate sucrose and glucose efflux [12–14], Tal2g of *X. oryzae* pv. *oryzicola* promotes lesion expansion and bacteria exudation by inducing the expression of sulfate transporter gene [15], AvrHah1of *X. gardneri* indirectly stimulates the expression of a pectate lyase gene to promote the accumulation of apoplectic fluid [16], AvrBs3 of *X. euvesicatoria* causes cell hypertrophy through increasing expression of pepper *UPA20* [17], Tal8 of *X. translucens* promotes accumulation of ABA through induction of *NCED* in wheat [18], and PthA4 of *X. citri* ssp. *citri* (*Xcc*) induces hypertrophy and hyperplasia through induction of citrus *CsLOB1* [19–22].

During the host-pathogen arms race, plants have evolved several strategies to combat *Xanthomonas* TALEs through altering or deleting the S gene promoter regions containing the EBE, utilization of executor R genes that harbor the EBE in their promoter to initiate immune response upon their induction, and recognition through NB-LRR resistance genes [23–31]. In return, *Xanthomonas* bacteria avoid these strategies by evolving different TALEs that target

different EBEs in the S gene promoter, to target a different or functionally similar S gene, and employing interference TALEs to suppress the TALE recognition by NB-LRR [4, 27, 32]. Employment of alternative TALEs directed to the same target was reported in at least two pathosystems. *Xoo* strains utilize multiple TALEs (TalC, PthXo3, Tal5, and AvrXa7) to target at least three independent EBEs in the promoter of *OsSWEET14* and use two other TALEs (PthXo1, and PthXo2) to induce the expression of *OsSWEET11* and *OsSWEET13* [12, 13, 33–35]. In addition, different EBEs in the promoter region of citrus *CsLOB1* were identified to be targeted by TALEs from *Xcc* (PthA4/PthA\*/PthAw2) [36] and *X. citri* ssp. *aurantifolii* (*Xca*) (PthB/PthC) [19].

Mutation of the EBE of S genes via TALEN and CRISPR mediated genome editing and utilization of naturally occurring EBE variants have been used to generate disease resistant crops, e.g., rice and citrus [37–42]. However, the tandem repeat nature of TALEs subjects them to high frequency of mutations and rearrangements [43], thus undermining the durability of resistant crops generated via mutating EBEs. It is pivotal to investigate how TALEs of pathogens adapt to the EBEs of S genes to develop successful strategies to breed or design durable disease resistance in crops.

*Xanthomonas* bacteria are highly specialized with narrow host range [5]. Like many other specialist pathogens, the mechanisms that dictate host specificity and adaptation are not fully understood. Investigations of host adaptation have been conducted by analyzing bacterial population genetics, reverse genetics studies or simulating host adaptation using experimental evolution. Evolutionary events, such as acquisition of novel pathogenicity associated gene clusters by horizontal genet transfer, altered regulation of metabolic genes, alteration or loss of genes associated with immune recognition by the host, and modification of existing virulence genes, were reported in host adaptation studies. For instance, acquisition of genes associated with detoxification of plant antimicrobial compounds was found to expand the host range of *Enterobacteria* plant pathogens *Pectobacterium* and *Panotea* to *Brassicales* and *Allium*, respectively [44, 45]. Alterations in the flg22-elicitor region in the flagella of *Ralstonia solanacearum* and *Xanthomonas oryzae* prevent the recognition by respective hosts [46, 47]. Field introduction of pepper and tomato lines bred with R genes against specific T3SS effectors of *Xanthomonas euvesicartoria* was followed by bacterial adaptation through disruption or modification of the targeted effectors and introduction of pathogen races that lack the corresponding effectors [48]. Experimental evolution approaches have been utilized as a tool to study host adaptation in animal and plant pathogens. Numerous studies have identified specific adaptive mutations that were involved in pathogenicity. For incidence, *Pseudomonas aeruginosa* experimentally evolved in mice exhibited missense mutations in the two-component sensor *pmrB* that regulates attachment, LPS and resistance to amicrobial compounds [49, 50]. *Ralstonia solanacearum* strains that experimentally evolved on bean plants harbored a mutation in the transcriptional regulator *efpR*, which regulates EPS production, motility and numerous metabolic processes [51, 52]. *Xcc* strains that evolved in resistant Meiwa kumquat via repeated inoculation and isolation harbored point mutations in the *pthA4* TALE that was later verified to be associated with elicitation of immune responses [53, 54].

Experimental evolution studies of host-pathogen interactions usually focus on utilizing the experimental system as a tool for gene discovery and less on the mutational events of specific virulence factors that occur during adaptations. It remains unknown whether *Xanthomonas* can overcome the resistance or loss-of-susceptibility owing to the incompatible interactions between TALEs and the EBE of the corresponding susceptibility genes. We hypothesized that TALEs have the potential to overcome the mismatches in the EBE of susceptibility genes and the adaptation capacity inversely correlates with the number of mismatches. To test this hypothesis, we utilized the *Xcc*–citrus pathosystem [55] as a model to investigate TALE

adaptations in overcoming incompatible interactions by using an experimental evolution approach. Indeed, our data provide strategic guidance for development of durable EBE-based resistance against TALE-dependent pathogens.

## Results

### Natural variations of citrus *LOB1* and TALEs in *Xanthomonas citri* suggest host adaptation

We investigated the variations among TALEs (PthA4 and homologs) that target *LOB1* by analyzing all available *Xcc* and *Xca* deposits in the NCBI database. We identified TALEs that display moderate to high binding affinity to the sweet orange *LOB1* promoter according to target finder feature of "TAL Effector Nucleotide Targeter 2.0" [56]. The analysis identified 20 *LOB1*-targeting TALEs (**Table 1**) that contain 13 unique repeat array variants (named RVDV1-RVDV13, **Table 1** and **Fig 1A**). The majority of the TALEs were represented by two dominant repeat array variants, RVDV1 and RVDV5. RVDV5 was identified in multiple *Xcc* genomes and represented by a single allelic variant. In addition, all *Xcc* strains containing RVDV5 were isolated from key lime or lemon trees in Florida (**Table 1**). On the other hand, RVDV1 was identified in six allelic variants and found in *Xcc* strains isolated from multiple hosts in numerous geographic regions (**Table 1**).

We assessed the phylogenetic and functional lineage of the *LOB1* targeting TALEs using the QueTAL tool [57] (**S1 Fig**). The analysis identified at least two independent subgroups within the *LOB1* targeting TALEs (**S1A Fig**). The first group, composed of RVDV12 and RVDV13, represented TALEs isolated from *Xca* strains in South America [58] (**Table 1** and **S1A Fig**). In addition to harboring a different repeat array composition, these two TALEs also potentially target a different EBE in the *LOB1* promoter, which only partially overlap with the EBE targeted by the other TALEs (**Fig 1A**). The second group, composed of RVDV5, RVDV6 and RVDV7, represented isolates of the lime-restricted *Xcc*[AW] found in North America [59] (**Table 1** and **S1A Fig**). While functional lineage analysis based on predicted EBE binding forecast different affinities from the rest of the TALEs (**S1B Fig**), genome based analysis found that these three TALEs target an identical EBE in the *LOB1* promoter to that of the other *Xcc* TALEs (**Fig 1A**) by utilizing different repeat arrays to target the same DNA sequence (**Fig 1A**). Even though the remaining TALEs share repeat stretches and high functional lineage between them (**Figs 1A and S1B**), distance analysis did not identify clear phylogenetic lineage (**S1A Fig**). It is unclear whether these TALEs were acquired or evolved independently of each other.

To investigate the relationship between the *LOB1* EBEs and *LOB1* targeting TALEs, we analyzed the sequences surrounding the EBEs in the *LOB1* promoter regions (p*LOB1*) of multiple *Rutaceae* plants including both citrus and non-citrus (**Table 2**). *LOB1* promoters were derived from available sequence deposits (https://www.citrusgenomedb.org/) or newly sequenced here (**Table 2**). We identified seven allelic variants in the *LOB1* promoter (named A to G, **Table 2**). The majority of commercial citrus genotypes contained at least one A allele, which presumably originated from the ancestral species mandarin orange (*C. reticulate*) [60] (**Table 2**).

Sequence analyses revealed that the 18 bp EBE[PthA4] [19] of p*LOB1* is 100% conserved in all commercial citrus cultivars (variants A, B and C, **Fig 1B**). However, we identified some sequence variations in the p*LOB1* of wild *Rutaceae* species (variants E, F and G, **Fig 1B**) and in the rootstock species Carrizo, Swingle citrumelo and Sour orange (variants D and E, **Fig 1B**).

The affinity of each of the *Xcc* TALE repeat array variants to the p*LOB1* variants was estimated using target finder feature of "TAL Effector Nucleotide Targeter 2.0"[56]. The analysis identified different specificity of the TALEs to specific promoter variants (**Fig 1C**). For instance, RVDV1 displayed high affinity to p*LOB1* variants A, B and C that are present in all

**Table 1. Natural variations among TALEs targeting *CsLOB1*.**

| RVD variant | RVD | Allelic variant | Bacteria | NCBI GenBank | Host | Geographic origin |
|---|---|---|---|---|---|---|
| 1 | NI N* NI NI NI HD HD NG HD NG NG NG NG NS HD HD NG NG | 1A | *Xcc* strains: 306, 306A, 5208, BL18, FB19, gd3, jx4, jx5, mf20, MN10, MN11, MN12, NT17, UI6, UI7, 03-1638-1-1 | AAM39311, AJD66579, AJZ37799, AJZ33330, AJZ28866, AJZ24451, AJZ20025, AJZ15601, AJZ11172, AJZ06700, AJZ02279, AJY97855, AJY93431, AJY88957, AJY84537, AJY80115, AUZ53767 | *Citrus sinensis* (Sweet Orange), *C. aurantifolia* (Key lime), *C. paradisi* (Grapefruit) | Brazil: São Paulo, USA: Florida, China: Guangdong, China: Jiangxi, Argentina |
| | | 1B | *X. citri* α strain NI-1 | BAA37119 | *C. natsudaidai* (Amanatsu) | Japan |
| | | 1C | *X. citri* α,b | WP_082243722 | *C. sinensis* (Sweet Orange) | China: Jiangxi |
| | | 1D | *Xcc* strains: LL074-4, LM180 | APR13430, OLR69148 | *C. paradise* (Grapefruit) | Martinique, Argentina |
| | | 1E | *Xcc* strain LH201 | APR27435 | *C. hystrix* (Kaffir lime) | Reunion |
| | | 1F | *Xcc* strain KC21 | BAF46271 | *C. grandis* (Pomelo) | Japan |
| 2 | NI NG NI HD NI HD HD NG HD NG NG NG NG NS HD NS NG NG NG | 2A | *Xcc* strain TX160149 | ARR15471 | *C. aurantifolia* (Key lime) | USA: Texas |
| 3 | NI NG NI NI NI HD HD NG HD NG NG NG NG NG NG HD NG NG | 3A | *X. citri* α strain XW47 | ACZ62652 | *C. paradise* (Grapefruit) | Republic of China: Taiwan |
| 4 | NI NI NI HD HD NG HD NG NG NG NG NS HD HD HD NG | 4A | *Xcc* strain Xcc049 | AHB33738 | *C. sinensis* (Sweet Orange) | China: Chong Qing |
| 5 | NI NG NG NG NS HD HD NS HD NG NG NG NG NS HD HD NG NG | 5A | *Xcc* strains: Aw12879, AW13, AW14, AW15, AW16 | AGI10546, AJZ64238, AJZ51443, AJZ46823, AJZ42208 | *C. aurantifolia* (Key lime), *C. limon* (Lemon) | USA: Florida |
| 6 | NI NG NG NG NS HD HD NS HD NG NC NG NG NS HD HD NG NG | 6A | *Xcc* strain X0053 | ABO77779 | *C. aurantifolia* (Key lime) | USA: Florida |
| 7 | NI NG NG NG NS HD HD NS HD NG NG NG NG NS HD HD NG NG NG | 7A | *Xcc* strains: TX160042, TX160197 | ARR19110, ARR20875 | *C. aurantifolia* (Key lime), *C. hystrix* (Kaffir lime) | USA: Texas |
| 8 | NI N* NI NI NI HD HD NG HD NG NG NG NG NS HD HD HD NG NG | 8A | *Xcc* strain Xcc29-1 | AYL23296 | Citrus[C] | China: Jiangxi |
| | | 8B | *Xcc* strain Xcc29-1 | AGH79796 | Citrus[C] | China |
| 9 | NI N* NI NI NI NG HD NG HD NG NG NG NG NS HD HD NG NG | 9A | *X. citri* α strain 3213 | AAC43587 | *C. paradise* (Grapefruit) | USA: Florida |

(*Continued*)

**Table 1.** (Continued)

| RVD variant | RVD | Allelic variant | Bacteria | NCBI GenBank | Host | Geographic origin |
|---|---|---|---|---|---|---|
| 10 | HD N* NI NI NI HD HD NG HD NG NG NG NG NS HD HD NG NG | 10A | *Xcc* strain LM180 | OLR69303 | *C. paradise* (Grapefruit) | Argentina |
| 11 | NI N* NI NI NI HD ND NG HD NG NG NG NG NS HD HD HD ND NG | 11A | *Xcc* strain Xcc49 | AYL27693 | Citrus[C] | China: Chongqing |
| 12 | HD NG HD NG NI NG HD NG HD NI NI HD HD HD HD NG NG NG | 12A | *X. citri* ssp. *aurantifolii* strain B69, *X. citri* [α, b] | WP_011153905, NP_942641, AAO72098 | Citrus[C] | South America |
| 13 | HD NG HD HD NI NG NI NG NI NI HD NG HD HD HD NG NG NG | 13A | *X. citri* ssp. *aurantifolii* strain ICPB 10535 | WP_088370900, EFF47385 | *C. aurantifolia* (Key lime) | Brazil: São Paulo |
| | | 13B | *X. citri* ssp. *aurantifolii* strain C340 | ABO77782 | *C. aurantifolia* (Key lime) | Brazil: São Paulo |

[α] *Xanthomonas* ssp. is not specified in deposit or the corresponding publication.

[b] strain is not specified in deposit or the corresponding publication.

[C] Citrus species is not specified in deposit.

Allelic variant means that the backbone is not identical, but the repeat array is identical.

commercial citrus varieties but only showed moderate affinity to p*LOB1* variants D, E, F and G, that are present in non-citrus *Rutaceae* species and rootstock varieties (**Fig 1C**). On the other hand, RVDV8, RVDV9, RVDV11 and RVD13 displayed only moderate affinity to the p*LOB1* variants found in most commercial citrus varieties but higher affinity to the EBE found in Carrizo citrange, Swingle citrumelo, *Poncirus trifoliate*, *C. aurantium* or *Ichang papeda* (**Fig 1C**). Our analyses suggest that p*LOB1*-targeting TALEs of *X. citri* evolved different specificity to *Rutaceae* hosts during host adaptation. The prevalence of RVDV1 in the *Xcc* populations is probably due to its high affinity to the widely presented EBEs (A, B and C) in the commercial varieties.

We validated the predicated promoter binding affinity *in vivo* by fusing p*LOB1* from sweet orange and Swingle citrumelo (variants A and E, respectively) to a GUS reporter (**Fig 1D and 1E**). The promoter activity was tested in the presence of p*LOB1*-targeting TALE PthA4 (RVDV1) using *Agrobacterium* mediated transient expression in *Nicotiana benthamiana* leaves. Consistent with the *in silico* prediction, PthA4 promoted significantly higher induction of sweet orange p*LOB1* than that of Swingle citrumelo (**Fig 1E**).

## Adaptation of p*LOB1*-targeting TALEs

In order to optimize EBE-mutating design to generate resistant varieties, we investigated how TALEs adapt to their corresponding EBE. The sweet orange-*Xcc* pathosystem was used to experimentally simulate TALE adaptation in overcoming incompatible interactions. To this aim we constructed eight designer TALEs (dTALEs) that harbored repeat arrays with different compatibilities to a 19 bp EBE in p*LOB1* of sweet orange (**Fig 2A**). First we constructed a PthA4-mimicking dTALE with a repeat array that perfectly matches the 19 bp EBE$_{PthA4}$ in p*LOB1* (dTALEWTLOB1, **Fig 2A**) and demonstrated it complemented a *Xcc pthA4* Tn5

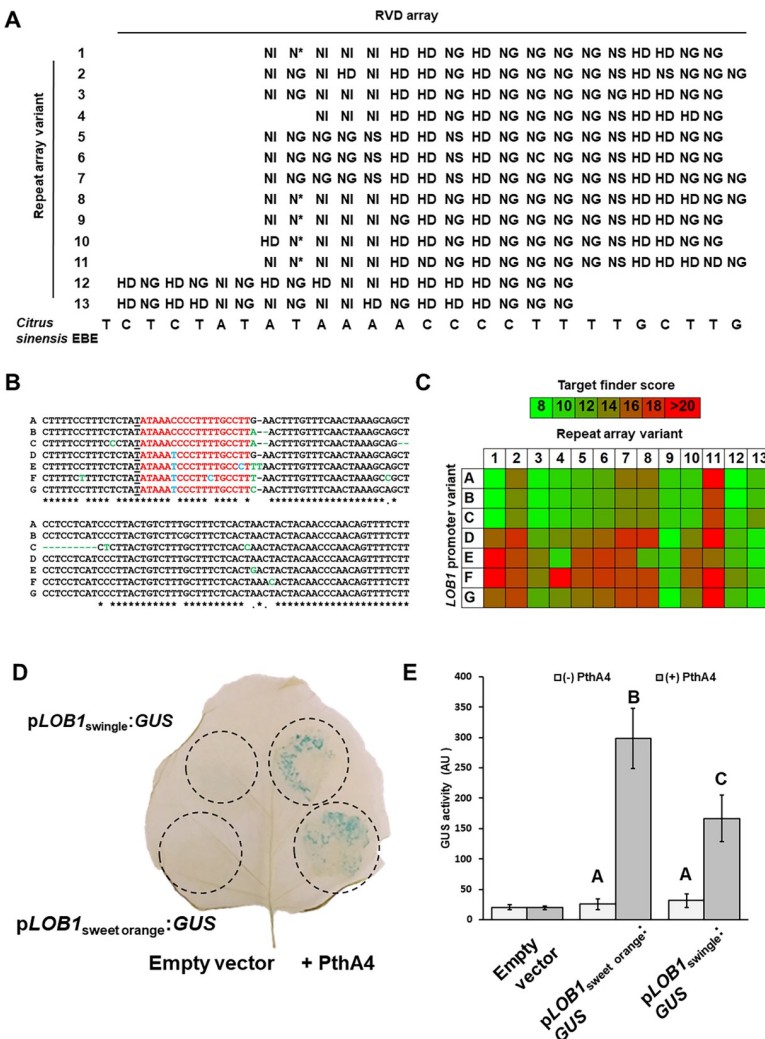

**Fig 1. Variability of *X. citri LOB1* targeting TALEs and the *LOB1* EBE region in *Rutaceae* species.** A. RVD repeat arrays of *LOB1* targeting TALEs from *X. citri* species (Sources are elaborated in Table 1). B. Sequence alignment of allelic variants (Sources are elaborated in Table 2) of the surrounding region of the TALE effector-binding elements (EBEs) from *Rutaceae* plants. Sequence alignment was conducted with Clustal Omega Multiple Sequence Alignment feature (https://www.ebi.ac.uk/Tools/msa/clustalo/) using default settings. Conserved residues in EBE region are marked in red. Variations in the EBE compared to allelic variant A are marked in blue. Variations in the area outside of the EBE compared to allelic variant A are marked in green. Thymidine residues proceeding EBEs are underlined. C. Target finding scores (lower scores indicate higher predicted binding affinity) of *LOB1* targeting TALEs against allelic variants of *Rutaceae LOB1* promoter according to TAL Effector Nucleotide Targeter 2.0 using Target Finder tool (https://tale-nt.cac.cornell.edu/). Scores are depicted in colored heat maps correlating to the ruler placed on the top of the table. D and E. Induced expression of sweet orange and Swingle citrumelo *LOB1* by PthA4. *Nicotiana benthamiana* leaves were inoculated with *Agrobacterium* to co-express His-PthA4 or an empty vector with GUS reporter under the control of the *LOB1* promoter from sweet orange (*Citrus sinensis*) or Swingle citrumelo (*Poncirus trifoliata* x *Citrus paradisi*). Expression of His-PthA4 was driven by an estradiol-inducible system and 17β-estradiol was applied at 24 h after agro-infiltration. D. Histochemical GUS staining of inoculated leave at 72 h after 17β-estradiol treatment. Experiment was repeated three times with similar results. E. GUS activity (arbitrary units [AU]) in inoculated areas was determined at 72 h after 17β-estradiol treatment. Values are means ± SE of nine biological replicates. The experiment was conducted three times and each experiment was composed of three biological replicates. Letters denote significant differences based on analysis of variance (Anova) and comparisons for all pairs using Student's *t*-test (*P*-value < 0.05).

Table 2. Variants in the *LOB1* promoter among *Rutaceae* species.

| Common name | Species/Genotype | LOB1 promoter variant | | | | | | | Comments |
|---|---|---|---|---|---|---|---|---|---|
| | | A | B | C | D | E | F | G | |
| Mandarin orange* | *Citrus reticulata* | √ | | | | | | | Ancestral species |
| Pomelo* | *C. maxima* [(Burm.) Merr], *C. grandis* Swingle, Tanaka | | √ | | | | | | |
| Citron* | *C. medica* | | | √ | | | | | |
| Sweet orange+,* | *C. sinensis* (*C. maxima* × *C. reticulata*) | √ | √ | | | | | | Commercial hybrid species |
| Grapefruit+ | *C. paradisi* (*C. maxima* × sweet orange) | √ | √ | | | | | | |
| Lemon+ | *C. limon* (sour orange × citron) | | √ | √ | | | | | |
| Mexican lime+ | *C. aurantiifolia* (micrantha x citron) | | √ | √ | | | | | |
| Clementine* | *C. clementina* (Willowleaf mandarin × sweet orange) | √ | | | | | | | |
| Sugar belle mandarin+ | "Clementine" mandarin × "Minneola" tangelo | √ | | | | | | | |
| Alemow+ | *C. macrophylla* [citron × biasong (*C. micrantha*)] | | | √ | | | | | Rootstock species |
| Sour orange+ | *C. aurantium* (*C. maxima* x *C. reticulata*) | √ | | | √ | | | | |
| Swingle citrumelo+ | *C. paradisi* × *Poncirus trifoliata* | √ | | | | √ | | | |
| Carrizo+ | *C. sinensis* × *Poncirus trifoliata* | √ | | | | √ | | | |
| Hong Kong kumquat* | *Fortunella hindsii* | √ | | | | | | | Wild species |
| Meiwa kumquat+ | *Fortunella crassifolia* | √ | | | | | | | |
| Trifoliate orange+,* | *Poncirus trifoliata* | | | | | √ | | | |
| Chinese box orange* | *Severinia buxifolia* | | | | | | √ | | |
| Papeda* | *Ichang papeda* | | | | | | | √ | |

*Information is based on sequence from www.citrusgenomedb or http://citrus.hzau.edu.cn/orange.

+information is based on amplification from genomic DNA and sequencing.

insertion mutant (*Xcc pthA4*:Tn5) in inducing *CsLOB1* expression and promoting canker symptoms (S2 Fig) [61]. We then constructed seven dTALEs with 2 to 9 mismatches of RVDs within their repeat arrays and tested their ability to complement *Xcc pthA4*:Tn5. DNA sequences of all the constructed dTALEs are available in S1 Text. As expected, the manufactured dTALEs (named dTALELBM1 to dTALELBM7, Fig 2A) did not complement *Xcc pthA4*:Tn5 and were unable to induce the expression of *CsLOB1* and *Xcc pthA4*:Tn5 carrying the dTALEs had incompatible interactions with the citrus host (S2 Fig).

Duplicates of *Xcc pthA4*:Tn5 carrying each of the seven dTALEs were subjected to *in planta* experimental evolution assays. *Xcc pthA4*:Tn5 carrying the dTALEs were inoculated into and reisolated from sweet orange leaves for 30 infection cycles, representing approximately 1,093 bacterial generations. Five of the 14 bacterial strains were able to induce canker symptoms in sweet orange within 9–28 cycles (Table 3) and dTALEs isolated from the five adapted strains were able to complement *Xcc pthA4*:Tn5 in inducing *CsLOB1* expression, causing canker symptoms, and promoting bacterial growth in sweet orange (Fig 3).

Sweet orange leaves inoculated with *Xcc pthA4*:Tn5 harboring dTALEs isolated from the adapted *Xcc* strains or dTALEWTLOB1 displayed canker symptoms between 4–7 days after inoculation (dpi) while leaves inoculated with *Xcc pthA4*:Tn5 or *Xcc pthA4*:Tn5 harboring the non-adapted dTALEs failed to cause canker symptoms after 14 days (Fig 3A). The ability to induce the expression of *CsLOB1* by the adapted TALEs was monitored at 36 and 72 hours post inoculation (hpi). The expression of *CsLOB1* in sweet orange was significantly increased by *Xcc pthA4*:Tn5 harboring the adapted dTALEs, i.e., dTALELB2A1, dTALELB2A2, dTALELB3A, dTALELB5A, and dTALELB7A, whereas the expression was not significantly altered by the original dTALEs (Fig 3B). In addition, introduction of dTALEWTLOB1 or the adapted TALEs to *Xcc pthA4*:Tn5 significantly improved bacterial colonization of sweet orange leaves,

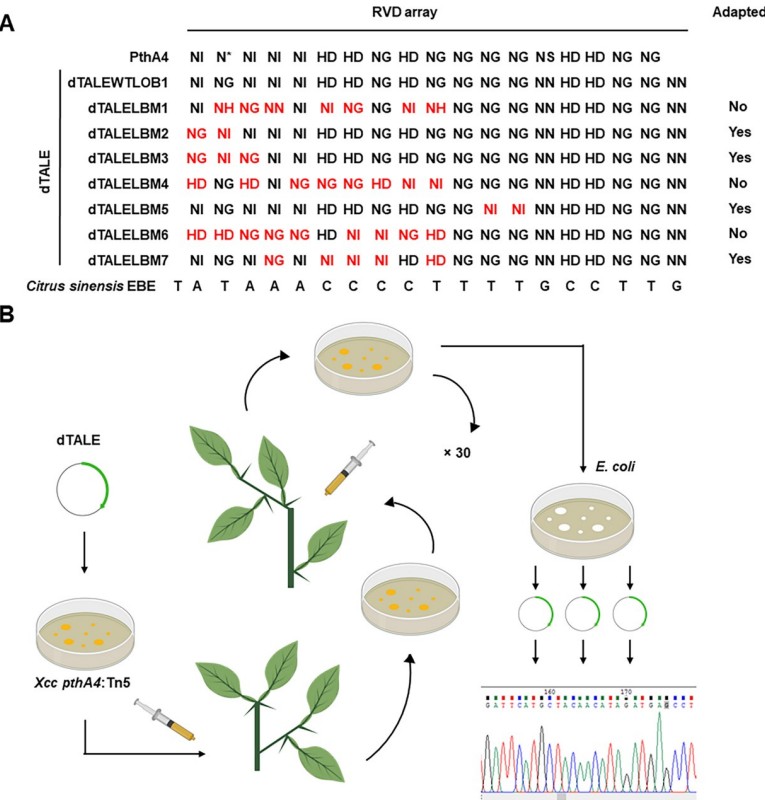

**Fig 2. Experimental evolution of TALEs.** A. RVD repeat arrays of PthA4 (XACb0065) and dTALEs used in experimental evolution test. The nucleotide sequence of the effector-binding element of *CsLOB1* from sweet orange (*Citrus sinensis*) is represented at the bottom. "Adapted" column indicates whether the dTALE variant was able to adapt in the duration of the experiment. B. Schematic representation of the experimental evolution workflow. Scheme was created with Biorender (https://biorender.com/).

reaching similar levels as the wild type *Xcc* at 12 dpi, whereas the four original dTALEs grew similarly as *Xcc pthA4*:Tn5 (**Fig 3C**).

Those five adapted strains corresponded to dTALELBM2, dTALELBM3, dTALELBM5, and dTALELBM7, which contain 2, 3, 2, and 5 mismatches, respectively. During this period, *Xcc pthA4*:Tn5 strains harboring dTALELBM1, dTALELBM4, and dTALELBM9 that contain at least 7 mismatches with EBE$_{PthA4}$ did not adapt to sweet orange.

As a negative control, *Xcc pthA4*:Tn5 carrying each of the seven dTALEs were streaked on artificial NA medium in parallel to the plant infection cycles to assess the effect of the selective pressure of incompatible plant environment on TALE adaptation. Plasmids were extracted from three single colonies of each of the seven strains after 30 streaking cycles and the DNA sequence of their repeat arrays were determined. We did not observe any modifications in the repeat arrays of TALEs adapted on NA medium and sequences were identical to the original non-adapted parental TALEs.

## Adapted TALEs display mutations and deletions in their repeat arrays

The sequence of the repeat region of TALEs was determined at cycle 30 for the 14 strains (**S1 Text**). The adapted variants isolated from strains that displayed canker symptoms and induced *CsLOB1* expression were sequenced at two time points, at the first sign of host adaptation (i.e., showing canker symptoms) and at the end of the experiment after 30 infection cycles along

**Table 3. RVD variants of the original and adapted dTALEs.**

| dTALE | dTALE RVD | Parental dTALE | Binding affinity score to the *LOB1* promoter[A] | | Number of infection cycles for adaptation | Found after 30 infection cycles[C] | Detected in replicate[D] | |
|---|---|---|---|---|---|---|---|---|
| | | | Score[B] | Best possible score[B] | | | 1 | 2 |
| dTALEWTLOB1 | NI NG NI NI NI HD HD NG HD NG NG NG NG NN HD HD NG NG NN | NA[E] | 7.35 | 5.34 | NA | NA | NA | |
| dTALELBM1 | NI NH NG NN NI NI NG NG NI NH NG NG NG NN HD HD NG NG NN | NA | -[F] | 5.27 | NA | YES | YES | YES |
| dTALELBM2 | NG NI NI NI HD HD NG HD NG NG NG NG NN HD HD NG NG NN | NA | 12.87 | 5.34 | NA | NO | NO | NO |
| dTALELB2A1 | NG NG NI NI NI HD HD NG HD NG NG NG NG NN HD HD NG NG NN | dTALELBM2 | 9.45 | 5.45 | ~17 | YES | YES | YES |
| dTALELB2A2 | NI NG NI NI NI HD HD NG HD NG NG NG NG NN | dTALELBM2 | 5.65 | 3.65 | ~17 | NO | NO | YES |
| dTALELBM3 | NG NI NG NI NI HD HD NG HD NG NG NG NG NN HD HD NG NG NN | NA | 14.97 | 5.45 | NA | YES | NO | YES |
| dTALELB3A | NG NI NI HD HD NG HD NG NG NG NG NN HD HD NG NG NN | dTALELBM3 | 9 | 5 | ~9 | YES | YES | NO |
| dTALELBM4 | HD NG HD NI NG NG NG HD NI NI NG NG NG NN HD HD NG NG NN | NA | 21.66 | 5.45 | NA | YES | YES | YES |
| dTALELBM5 | NI NG NI NI NI HD HD NG HD NG NG NI NI NN HD HD NG NG NN | NA | 14.17 | 5.14 | NA | YES | YES | NO |
| dTALELB5A | NI NG NI NI NI HD HD NG HD NG NI NG NN HD HD NG NG NN | dTALELBM5 | 10.76 | 5.24 | ~19 | YES | NO | YES |
| dTALELBM6 | HD HD NG NG NG HD NI NI NG HD NG NG NG NN HD HD NG NG NN | NA | 25.66 | 6.81 | NA | YES | YES | YES |
| dTALELBM7 | NI NG NI NG NI NI NI NI HD HD NG NG NG NG NN HD HD NG NG NN | NA | 16.63 | 5.26 | NA | YES | NO | YES |
| dTALELB7A | NI NG NI NG NI NI NI HD HD HD HD HD NG NG NN | dTALELBM7 | 8.86 | 3.58 | ~28 | YES | YES | NO |

Note

[A] The binding affinity analysis was conducted in *LOB1* variant A from sweet orange (*Citrus × sinensis*). The promoter region was set as the 1,000 bp sequence upstream of the transcriptional start site.

[B] According to the target finder tool provided by https://tale-nt.cac.cornell.edu/.

[C] A clone is defined as "detected" if the dTALEs or adapted TALEs were present in plasmids isolated from bacteria at cycle 30. Three independent clones per strain were isolated from *Xcc pthA4*:Tn5, introduced to *E. coli* and sequenced.

[D] Each experiment was conducted with two replicates marked as "1" and "2". Data states whether the inducted dTALE was identified in each replicate in the duration of the experiment representing both the time of adaptation and the end of the experiment as cycle 30.

[E] NA: not applicable.

[F] Score is beyond cutoff.

Blue color indicates adapted TALEs.

with the rest of the strains. The repeat arrays of TALEs extracted from the strains that were unable to promote canker at cycle 30 were identical to their parental dTALEs (**Table 3**). The other five adapted TALE variants, which were able to complement *Xcc pthA4*:Tn5 (**Fig 4**), contained alterations in the repeat arrays compared to the parental dTALEs (**Table 3**).

The two adapted dTALE variants of dTALELBM2 (dTALELB2A1 and dTALELB2A2) that contained two mismatches in the first two repeats, displayed distinct repeat rearrangements: the first adapted variant, dTALELB2A1, was identified in both duplicate strains after 17 infection cycles. In this variant, the RVD of the second repeat was changed from NI to NG, which matches the corresponding target "T" nucleotide in the EBE$_{PthA4}$ (**Fig 4A and 4B**, **Table 3**). The second adapted TALE, dTALELB2A2, was identified after 17 infection cycles in one of the duplicate strains. dTALELB2A2 contained mutations in 7 repeats: the first and second mismatched repeats were altered from NG-NI to NI-NG, matching the first "AT" target site in EBE$_{PthA4}$. In addition, we observed a deletion of the five C-terminal repeats. These mutations altered dTALELB2 from a TALE containing 19 repeats with two mismatches into a TALE with 14 repeats with a perfect matching repeat array (**Fig 4A and 4B**, **Table 3**).

Adaptation was observed in dTALELBM3, which originally contained mismatches in the first three repeats, in one of the duplicates after nine infection cycles and at the end of the experiment after 30 infection cycles. The adapted variant, dTALELB3A, displayed a deletion of

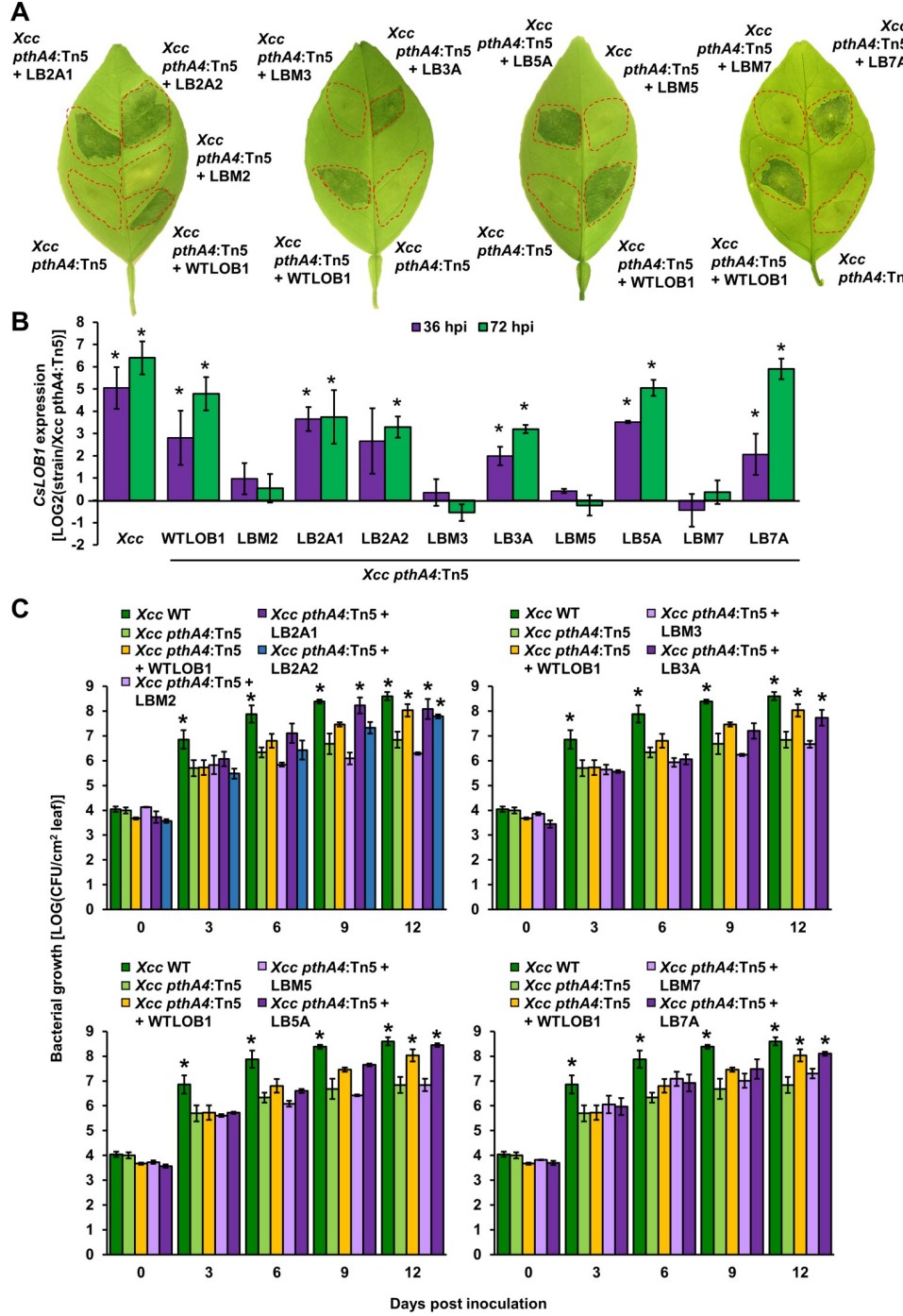

**Fig 3. Functional characterization of adapted dTALEs.** Sweet orange leaves were syringe-infiltrated with suspensions ($1 \times 10^8$ CFU/mL for A and B, $1 \times 10^6$ CFU/mL for C) of *Xcc* 306 (*Xcc* WT), *Xcc pthA4*:Tn5 or *Xcc pthA4*: Tn5 transformed with the parental and adapted dTALEs depicted in Fig 4A. A. Inoculated leaves were photographed at 7 days post inoculation. The experiments were repeated three times with similar results. B. The gene expression of *CsLOB1* was quantified at 36 and 72 h post inoculation (hpi) using quantitative reverse transcription PCR. The *GAPDH* gene was used as an endogenous control. Values are means ± SE of three biological replicates. C. Bacterial growth *in planta*. Values represent means ± SE of three biological replicates. The experiments were repeated three times with similar results. B and C. Asterisks indicate a significant difference (Student's *t*-test, *P*-value < 0.05) compared to *Xcc pthA4*:Tn5.

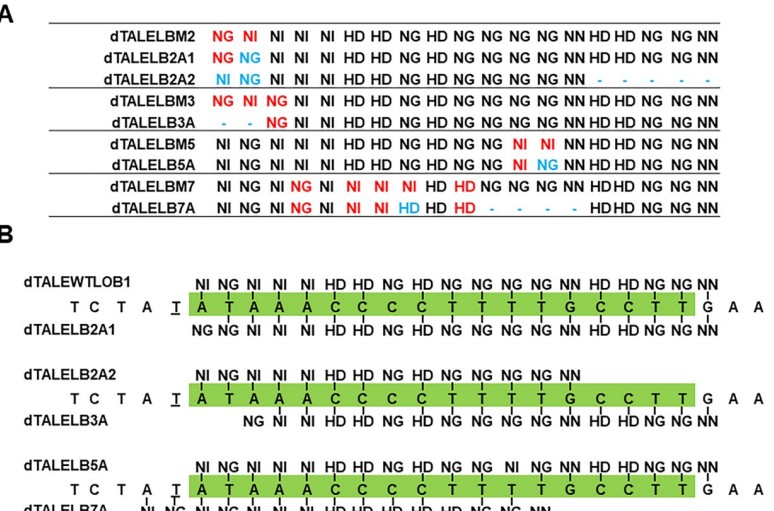

**Fig 4. Repeat rearrangements in adapted dTALEs.** A. RVD repeat arrays of parental and adapted dTALEs. Red-colored RVDs represent original mismatches compared to dTALEWTLOB1. Blue color indicates deleted or altered repeats in the adapted dTALEs compared to parental dTALEs. B. Predicted binding of adapted dTALEs [determined according to TAL Effector Nucleotide Targeter 2.0 using Target Finder tool (https://tale-nt.cac.cornell.edu/] to the *CsLOB1* in sweet orange (Chromosome 7, 28358599–28358574, allelic variant A) EBE. The PthA4 effector-binding element (EBE) is labeled in green and thymidine residue proceeding the EBE is underlined.

the first two mismatched repeats, altering dTALELBM3 from a 19 repeats TALE with three mismatches into a 17 repeats TALE with a single mismatch (**Fig 4A and 4B, Table 3**).

The adapted dTALELBM5 variant, dTALELB5A, was detected after 19 infection cycles and 30 cycles in one of the duplicate strains. dTALELB5A contained a NI to NG change in the mismatched repeat 12, which corresponds to the 12$^{th}$ "T" position in EBE$_{PthA4}$ (**Fig 4A and 4B, Table 3**).

Adaptation of dTALELBM7, which originally harbored five mismatched repeats, occurred only near the end of the experiment at the cycle 28 in one of the duplicates. The adaptive variant dTALELB7A displayed alteration of five repeats at positions 8–14 (**Fig 4A and 4B, Table 3**).

The full-length CDSs of the adapted dTALEs were sequenced. Other than repeat deletions or alterations of the RVDs, sequence analyses did not identify any other differences. Additionally, the altered nucleotides in the adapted RVDs displayed similar codons to building blocks encoding the same RVD in the dTALE repeat constructs, but different from the codons found in the native *Xcc* TALEs PthA1, PthA2, PthA3 and PthA4. This suggests that the repeat adaptations observed here probably occurred by recombination between the repeats within the dTALEs and not through point mutation nor recombination with the native TALEs of *Xcc*.

## Target analyses of adapted TALEs

After establishing the alterations in the adapted dTALEs, we further assessed their putative targets. We determined the potential promoter targets of the adapted dTALEs in sweet orange via *in silico* analyses. To this aim, we predicted the affinity of the parental and adapted dTALEs to the promoter sequences of all coding genes of sweet oranges (designated as 1 kb sequence upstream of the putative transcriptional start sites) using target finder feature in "TAL Effector Nucleotide Targeter 2.0"[56]. All adapted dTALEs demonstrated significantly higher affinity to the promoter sequence of *CsLOB1* (**S1 Data**, the *CsLOB1* gene is marked in green) than the

parental dTALEs. The predicted EBEs of the adapted dTALEs largely overlapped with the EBE$_{PthA4}$ (**Fig 4B**).

Additionally, our analysis predicted that some of the adapted dTALEs (dTALELB2A2, dTALELB3A and dTALELB7A. **S1J, S1K and S1M Data**) displayed relatively high affinity to promoters in addition to *pLOB1*. In particular, dTALELB7A was predicted to bind to several EBEs that are found in the proximity of the transcriptional start site of other genes than *CsLOB1* with similar or even stronger affinity (**S1M Data**). Among these genes, we identified several genes that encode proteins that are associated with canker development [19, 20], such as polygalacturonase (Cs2g27910) and sugar transporter (Cs9g05220) (**S1M Data**). It remains to be determined whether such adaptations play any roles in selection of the corresponding dTALEs.

## Discussion

Plant pathogenic bacteria usually possess high host specificity and most *Xanthomonas* species infect a very narrow range of hosts [5]. *Xanthomonas* host specificity is dictated by multiple factors, one of which is the induction of S genes by TALEs. Intriguingly, induction of the *CsLOB1* gene, the canker S gene, by *Xcc* PthA4 is essential for canker development, and consequently, the *Xcc pthA4* mutant is unable to cause canker symptoms [61]. Analyses of the *LOB1* promoter regions in various *Rutaceae* plants identified variations in the promoter sequences. However, the EBEs are completely conserved in the promoters identified in commercial citrus varieties and variations were only observed in non-citrus *Rutaceae* and rootstock varieties. This suggests that TALEs targeting *LOB1* promoters have adapted to their hosts by targeting a highly conserved region in the S gene promoter and by doing so efficiently enhanced the fitness of the pathogen. Consistent with this notion, RVDV1, which is the most abundant and geographically spread repeat array variant within the *Xcc* TALEs targeting *LOB1*, has the highest predicted binding affinity to the EBE of *LOB1* from commercial citrus varieties [21, 62].

This study provides experimental evidence that mutations and rearrangements of repeats of TALEs enable the adaptation of *Xanthomonas* on incompatible hosts. We observed adaptive mutations and rearrangements in five adapted TALEs from 14 independent events within a period of 9–28 infection cycles. In the adapted TALEs, mutations and rearrangements resulted in higher affinity to the EBE in the promoter of *CsLOB1*. *Xcc* bacteria carrying these TALEs were able to induce the expression of *CsLOB1* that caused citrus canker symptoms and enhanced leave colonization.

Erkes et al. 2017 characterized the adaptation events that occurred in *X. oryzea* TALEs using *in silico* techniques and genomic analysis [63]. This elegant study reported that changes in repeat arrays are mainly associated with repeat deletion, recombination with different repeat arrays of other TALEs and point mutations. Three of our adaptive variants displayed repeat deletions and four displayed substitution of the RVDs in specific repeats. The changes in the TALE repeat arrays probably resulted from the misalignment-mediated rearrangements, which are common for repetitive DNA sequences. One genetic hallmark of misalignment-mediated rearrangements is their independence of homologous recombination factors, including the RecA strand transfer protein of bacteria [43]. Multiple features of the tandem repeats of TALEs facilitate their adaptations since it has been suggested that the length, and proximity of the repeats are among the important determinants of their propensity to rearrange [43]. Tandem repeats of over a hundred nucleotides in length are deleted at very high rates, more reminiscent of recombination (10E-4) than of mutational (10E-8) frequencies [43]. In addition, there is an exponential dependence of deletion rate on proximity of the repeats [64], presumably because the two repeats must interact within a single replication fork. The tandem

repeats of TALEs fit both parameters for RecA-independent 'illegitimate' recombination [43, 65]. Although several other mechanisms can contribute, in theory, to tandem repeat mutations, it is plausible that most repeat mutations and rearrangements occur by misalignment during replication [66]. Additionally, the codon usage in the altered repeats matched the one used within the dTALEs (**S1 Text**), but not that of PthA1, PthA2, PthA3 and PthA4, indicating that these alterations are likely to originate from recombination within the introduced dTALE. Taken together, we infer that TALE adaptations result mostly from the RecA-independent 'illegitimate' recombination between repeats of the dTALE.

TALEs adaptations were only observed in dTALEs with less than seven mismatches from the target EBE of the *S* gene, providing useful information regarding how to modify the EBE-region for development of resistance against TALE-department pathogens and preventing or decelerating the resistance loss owing to TALE adaptations. Specifically, the five adaptive TALE variants originated from parental dTALEs that harbored between two to five mismatched repeats (*i.e.* dTALELBM2, dTALELBM3, dTALELBM5 and dTALELBM7), whereas non-adaptive TALEs were identified in the three dTALEs that harbored at least seven mismatched repeats (*i.e.* dTALELBM1, dTALELBM4 and dTALELBM6). The location of mismatches seems not to be a determinant factor of adaptations. Both dTALELBM2 and dTALELBM5 contained two tandem mismatches at the N-terminal and in the middle, respectively, and both underwent adaptations. The number of generations required for adaptation for the adapted TALE-containing *Xcc* stains was estimated to range from 328 to 1,020. We infer that the relatively short adaptation time results from the small number of recombination events needed for adaptations of dTALEs with 2–5 mismatches and the high recombination rate (10E-4) [43]. Three of the five adaptive TALEs can be enabled by a single recombination event (deletion of the first two repeats in dTALELB3A and a replacement of a single repeat in dTALELB2A1 and dTALELB5A, **S3 Fig**). On the other hand, the fourth adaptive variant, dTALELB2A2, contained a two-repeat replacement and a deletion of a five-repeat stretch, and the fifth adaptive variant, dTALELB7A, harbored a substitution and a deletion of four-repeat stretch, both of which can be achieved with as few as two recombination events (**S3 Fig**). However, when more mismatches (≥7) are present between TALEs and EBEs, it is probable that multiple recombination events are required to eliminate the mismatches, significantly reducing the possibility of generation of adaptive TALEs as observed for dTALELBM1, dTALELBM4 and dTALELBM6. Of note, we did not observe any changes in dTALEs isolated from non-adaptive variants. It is assumed that mutations occur to all constructs including dTALEs carrying seven or more mismatches. However, the probability for strains carrying less mismatches to overcome the mismatches via recombination and deletion is much higher than strains containing more mismatches. The mutated constructs that overcame the mismatches enable higher fitness for the strain, leading to takeover of the population. For the mutations that did not enable increased fitness for strains that carry the dTALEs containing more mismatches, the fact that they were not detected probably results from the extreme low percentage of such mutations in the population.

While our results clearly demonstrate an adaptive repeat rearrangement and deletion of various TALEs to overcome the mismatches, it is important to note that our study was conducted via an artificial experimental simulation rather than in natural settings. Our TALEs were cloned into pBBR1MCS5 [67], which is a medium copy number vector (estimated to be around 30 copies, [68]) while naturally occurring TALEs are encoded on low-copy mega plasmids or the bacterial chromosome. A recent survey by our group showed that the majority of *Xcc* strains contain three copies of plasmids (pXAC33 and pXAC64) in each bacterial cell. Thus, the experimental evolution using pBBR1MCS5 with higher copy number than the natural plasmid might expedite the mutation and selection process. In addition, since the

simulation was conducted in the greenhouse via syringe inoculation, it probably demonstrates the general feasibility of adaptation even though the kinetic and mechanism in a complex natural system might differ. First, during our experiment the passages had to go through the NA medium containing antibiotic selection between cycles. This procedure was a technical necessity to ensure culture purity. In natural settings, the bacteria will be subjected to more consistent selective pressure that would probably haste TALE adaptation kinetics or alternatively encourage TALE-independent adaptations to the host such as alteration in metabolic regulation or surface proteins profile [49, 51]. Second, syringe inoculations enable high titers of *Xcc* strains containing mismatching TALEs to establish *in planta*, which otherwise normally do not reach such high titers in natural settings. For example, the *pthA4* mutant of *Xcc* [61] can only establish very low titers via foliar spray that mimics the natural infection of *Xcc* compared with syringe inoculation. Consequently, our setting enables us to investigate the TALE adaption to overcome incompatible interactions, which is probably much rarer and slower in the natural settings. Third, we used a simplified closed system that eliminates factors including unstable environmental factors, competitive and mutualistic interactions with other microorganisms and interaction with different *Xanthomonas* strains that may lead to inter-bacterial recombination events [63]. Further work should be conducted to assess the ability of natural *Xanthomonas* strains to overcome miss-matched EBE of S genes in the field. Such work can utilize homozygous lines of citrus that were modified in the EBE of *LOB1* [42, 69] and examine the durability of field resistance to canker for extended time period and determine the putative adaption.

In summary, this study provides experimental evidence of TALE adaptations that convert incompatible to compatible interactions and offers guidance regarding how to potentially overcome the resistance loss due to TALE adaptations. Mutation of EBEs via TALEN or CRISPR-based genome editing and utilization of naturally occurring EBE variants have been regarded as one of the most efficient approaches to breed or develop resistant varieties against TALEs-containing pathogens [33, 37, 38, 70]. Our data suggest that mutation multiple nucleotides in the EBEs might be required to empower durable host resistance against TALE-dependent pathogens.

## Materials and methods

### Bacterial strains and plasmids

The bacterial strains and plasmids used in this study are listed in **S1 Table**. Oligonucleotides used for cloning and sequencing in this study are listed in **S2 Table**. *Xanthomonas citri* was grown at 28˚C in nutrient broth (NB) medium (Beef extract 3 g/L, Peptone 5 g/L) and on nutrient agar (NA) plates. *E. coli* and *A. tumefaciens* were grown in Luria-Bertani (LB) medium at 37˚C or 28˚C, respectively. When required, growth media were supplemented with gentamicin (5 μg/mL), kanamycin (50 μg/mL), tetracycline (5 μg/mL), ampicillin (100 μg/mL) and spectinomycin (100 μg/mL).

### Analysis of *Rutaceae LOB1* promoters, *Xanthomonas citri* TALEs and EBE affinity predictions

Genomic DNA was extracted from fully expanded leaves of various *Rutaceae* species (**Table 2**) using NucleoSpin Plant II (TaKaRa Bio Inc. Kusatsu, Japan). The *LOB1* promoter regions containing the PthA4 EBE were amplified from genomic DNA using Q5 High-Fidelity DNA Polymerase (NEB, Ipswich, MA) and fragments were cloned into pGEM-T vector (Promega, Madison, WI). DNA sequence was determined for 3–5 clones. Amplified *LOB1* promoter

sequences, along with *LOB1* promoter regions of other *Rutaceae* species available at the citrus genome database (https://www.citrusgenomedb.org/) were analyzed using the Clustal Omega multiple sequence alignment tool (https://www.ebi.ac.uk/Tools/msa/clustalo/) and separated into allelic variants.

TALE protein sequences of *X. citri* were extracted from NCBI protein database (https://www.ncbi.nlm.nih.gov/protein/?term=) and the compositions of RVDs in repeat arrays were manually determined. Binding affinity was analyzed against the promoter region of *LOB1* using target finder feature of "TAL Effector Nucleotide Targeter 2.0" [56] (parameters were set to score cutoff of 4.0, T only upstream base, and Doyle scoring matrix). All TALEs that were predicted to bind to *LOB1* according to score cutoff of 4.0 were considered as putative *LOB1* targeting TALEs.

## *Agrobacterium*-mediated transient expression and GUS activity measurements

For construction of transient expression vector of PthA4, *His-pthA4* was cloned from pET28-PthA4 [71] into pER8 [72]. For construction of β-Glucuronidase (*gus*) reporters the 913 bp *LOB1* promoter region was amplified from genomic DNA of sweet orange or Swingle citrumelo and cloned into p1380-35S-GUS [73], replacing 35S promoter. Binary vectors were transformed into *Agrobacterium* GV2260 by electroporation. *Agrobacterium* strains carrying GUS reporters and PthA4 constructs were co-infiltrated (OD600 = 0.1) into *Nicotiana benthamiana* leaves. Transient expression and *XVE* induction were conducted as previously described [74]. Histochemical staining of GUS was conducted as previously described [75]. For GUS activity measurements leaf disks of 1.5 cm diameter were collected at three days post *XVE* induction, homogenized in PBS (pH 7.0) and centrifuged at 14,000 rpm for 10 min at 4°C. Supernatants were analyzed for GUS activity as described elsewhere [76]. GUS activity was quantified by arbitrary units (AU) and determined as $1000 \times [A405 / (\text{time in min} \times \text{total protein in } \mu g \times 0.02)]$.

## Construction of designer TALEs

Designer TALEs (dTALEs) containing the repeat arrays elaborated in **Fig 3A** were constructed using "Golden Gate TALEN and TAL Effector Kit 2.0" as previously described [77] and cloned into pTAL2 as a final destination vector. The pTAL2 PstI/EcoRI fragments containing the dTALEs were cloned into pBBRNPth [54] and transformed into *Xcc pthA4*:Tn5 [61] by electroporation. Expression of all constructed dTALEs and their adapted derivatives in *Xcc* was validated by Western blot [78] using Anti-HA High Affinity antibody (Roche diagnostics, Basel, Switzerland) (**S4 Fig**).

## Plant inoculations, measurement of *CsLOB1* expression and measurement of bacterial growth

Bacteria were inoculated into expanded leaves of 2-year-old Valencia sweet orange plants with bacterial suspensions ($5 \times 10^5$ CFU/mL as initial inoculum in experimental evolution test, $10^6$ CFU/mL for monitoring bacterial growth and $10^8$ CFU/mL for monitoring symptom development and expression analysis of *CsLOB1*) in 10 mM MgCl$_2$ using a needless syringe. Plants were kept in a greenhouse at 28°C under natural light.

*CsLOB1* expression was measured in sweet orange leaves at 36 and 72 hours post bacterial inoculation. RNA isolation and qPCR analysis were conducted as described previously [76].

To measure bacterial growth *in planta* two leaf discs of 0.4-cm-diameter per plant from three plants were sampled, homogenized in 10 mM $MgCl_2$ and bacterial numbers were determined by plating 10 μL from 10-fold serial dilutions and counting the resulting colonies.

## Experimental evolution procedure

Two duplicate strains of *Xcc pthA4*:Tn5 carrying a vector encoding dTALELBM1, dTALELBM2, dTALELBM3, dTALELBM4, dTALELBM5, dTALELBM6 or dTALELBM7 were inoculated ($5 \times 10^5$ CFU/mL) into leaves of two independent sweet orange plants. Bacteria were isolated from leaves 7–10 days later from the two plants. Bacteria were plated on NA plates with gentamicin and kanamycin and bacterial populations were determined. Of note, we initially started the experiment using plant system alone but encountered many technical issues with contaminations. To overcome such issues, we added one isolation step to remove the contamination and guarantee the purity of the aforementioned *Xcc* strains.

Bacteria from each duplicate (two duplicate strains per dTALE–a total of 14 samples) were scrapped from NA plated, diluted to $5 \times 10^5$ CFU/mL and inoculated into leaves of two previously uninfected sweet orange plants. The procedure was repeated for 30 cycles, representing approximately 1,093 generations. Bacterial titers and appearance of canker symptoms were determined for each infection cycle. Generation time (G) was calculated as $G = T \times LOG_2 (B)$ where T represent the number of days and B represents the average daily growth rate of *Xcc pthA4*:Tn5 in sweet orange during exponential phase.

As a negative control, *Xcc pthA4*:Tn5 strains harboring the seven dTALEs used in the experimental evolution study were streaked on rich NA medium supplemented with gentamicin and kanamycin in parallel to the infection cycles to identify random occurrence of repeat rearrangement that is independent of host adaptation. Plasmids were extracted from three independent colonies of each of the NA streaked bacteria after 30 streaking cycles and sent for further analysis.

## Isolation, sequencing and validation of adapted dTALEs

Adapted dTALEs were extracted from *Xcc* following the first observation of canker symptoms in sweet orange leaves and at the end of the experiment (30 cycles). dTALE plasmids were extracted from *Xcc* using plasmid miniprep (ultra-fast): NucleoSpin Plasmid EasyPure kit (TaKaRa Bio Inc. Kusatsu, Japan) and transformed into *E. coli*. Plasmids were extracted from 5–10 colonies and introduced into *Xcc pthA4*:Tn5. Single colonies from each transformation were used for inoculation ($10^8$ CFU/mL) of sweet orange leaves. If an inoculation resulted in canker symptoms, the RVD compositions of the repeat array were determined by sequencing (Eton Bioscience, Inc., San Diego, CA). The sequence of adapted dTALEs (containing the TAL backbone and repeat arrays) was determined by sequencing. DNA sequences of the adapted dTALEs identified in this study are shown in **S1 Text**.

## Prediction of effector-binding elements

The 1 kb upstream sequences from the putative transcriptional start site of all genes in sweet orange were determined (**S2 Text**) and used as predicted promoters for affinity analyses. The affinity of dTALEs used in the study to sweet orange promoters was analyzed using target finder feature of "TAL Effector Nucleotide Targeter 2.0" [56] (parameters were set to score cutoff of 3.0, T only upstream nucleotide, and Doyle scoring matrix). The predicted EBEs are shown in **S1 Data**.

## Supporting information

**S1 Fig. Phylogenetic and functional relationships between *LOB1* targeting TALEs.** The RVD variants of *LOB1* targeting TALEs of *Xcc* and *Xca* (Table 1) were analyzed using QueTAL (http://bioinfo-web.mpl.ird.fr/cgi-bin2/quetal/quetal.cgi). A. Phylogenetic relationship between *LOB1* targeting TALEs was analyzed using DisTAL v1.1. B. Functional relationship between *LOB1* targeting TALEs was analyzed using FuncTAL v1.1.
(PDF)

**S2 Fig. Contribution of dTALEs to development of canker symptoms and expression of *CsLOB1*.** Sweet orange leaves were syringe-infiltrated with suspensions ($1 \times 10^8$ CFU/mL) of *Xcc pthA4*:Tn5 or *Xcc pthA4*:Tn5 transformed with the dTALEs depicted in Fig 2A. A. Inoculated leaves were photographed at 7 days post inoculation. B. The expression of *CsLOB1* was quantified at 96 h post inoculation. The *GAPDH* gene was used as an endogenous control. Values are means ± SE of three biological replicates. Asterisks indicate a significant difference (Student's *t*-test, *P*-value < 0.05) compared to *Xcc pthA4*:Tn5. The experiments were repeated three times with similar results.
(PDF)

**S3 Fig. Predicted recombination events that occurred in adapted dTALEs.** Schemes represent the alterations observed in the adapted dTALEs compared to their parental dTALEs and the predicted recombination events, which led to the adaptation. Repeats that were likely to be subjected for recombination or deletion in the parental dTALE are underlined. Repeats in the adapted dTALEs that were altered as a result of recombination are underlined and marked in blue. Repeats that were deleted are marked in purple. A. Alteration observed in dTALE2A1 compared with dTALELBM2. B. Alteration observed in dTALE2A2 compared with dTALELBM2. C. Alteration observed in dTALE3A compared with dTALELBM3. D. Alteration observed in dTALE5A compared with dTALELBM5. E. Alteration observed in dTALE7A compared with dTALELBM7.
(PDF)

**S4 Fig. Protein expression of dTALEs.** Total protein was extracted from overnight cultures of *Xcc pthA4*:Tn5 [No vector control (NVC)], *Xcc pthA4*:Tn5 carrying pBBR1MCS-5 [Empty vector (EV)] and *Xcc pthA4*:Tn5 transformed with the parental and adapted dTALEs. Samples were separated by SDS-PAGE and immunoblotted with the anti-HA antibody (upper panel) or stained with coomassie blue (lower panel).
(PDF)

**S1 Table. Bacterial strains and plasmids used in this study.**
(DOCX)

**S2 Table. Primers used in this study.**
(DOCX)

**S1 Text. dTALEs used in this study.**
(DOCX)

**S2 Text. The 1 kb upstream sequences from the putative transcriptional start site of all genes in sweet orange.**
(DOCX)

**S1 Data. Predicted sweet orange EBEs of TALEs used in this study.**
(XLSX)

## Acknowledgments

We would like to thank Jin Xu for his technical assistance.

## Author Contributions

**Conceptualization:** Doron Teper, Nian Wang.

**Data curation:** Doron Teper.

**Formal analysis:** Doron Teper.

**Funding acquisition:** Doron Teper, Nian Wang.

**Investigation:** Doron Teper.

**Methodology:** Doron Teper.

**Project administration:** Nian Wang.

**Resources:** Nian Wang.

**Software:** Doron Teper.

**Supervision:** Nian Wang.

**Validation:** Doron Teper.

**Visualization:** Doron Teper.

**Writing – original draft:** Doron Teper, Nian Wang.

**Writing – review & editing:** Doron Teper, Nian Wang.

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
