## [Decision Letter · Decision Letter 0]

21 Oct 2020

Dear Dr Wang,

Thank you very much for submitting your Research Article entitled 'Consequences of adaptation of TAL effectors on host resistance against Xanthomonas' to PLOS Genetics. Your manuscript was fully evaluated at the editorial level and by independent peer reviewers. The reviewers appreciated the attention to an important problem, but raised some substantial concerns about the current manuscript. Based on the reviews, we will not be able to accept this version of the manuscript, but we would be willing to review a much-revised version. We cannot, of course, promise publication at that time.

If you decide to revise the manuscript for further consideration at PLOS Genetics, please aim to resubmit within the next 60 days, unless it will take extra time to address the concerns of the reviewers, in which case we would appreciate an expected resubmission date by email to plosgenetics@plos.org.

[LINK]

We are sorry that we cannot be more positive about your manuscript at this stage. Please do not hesitate to contact us if you have any concerns or questions.

Yours sincerely,

Gregory P. Copenhaver

Editor-in-Chief

PLOS Genetics

Reviewer's Responses to Questions

**Comments to the Authors:**

Reviewer #1: This is a very interesting study on host adaption of a bacterial pathogen. The authors used a unique system (PthA4-LOB1 interaction in citrus canker disease) to show host adaption in Xanthomonas can be achieved by mutation of a specific TAL effector. The authors first examined the natural variations of PthA4 and LOB1 EBE sequence, which suggested they determine specificity of host-pathogen interaction in the disease system. The authors then designed pthA4 TALEs that had low binding affinity to LOB EBE, thus cannot promote disease after being introduced into the bacterium. Through many cycles of re-isolation and re-inoculation, some bacterial strains gained the ability to cause disease on the host and it was found that designed pthA TALEs in those strains mutated to a form with a stronger binding affinity to LOB EBE. This study provides the first experimental evidence that Xanthomonas pathogens can gain the ability to cause disease on a specific host by mutation of a single TALE. This may also give us an insight into molecular mechanisms of the evolution of bacterial pathogens, particularly for Xanthomonas. The experiments are well designed and executed. The manuscript is well-written and organized. The following is the list of comments and suggestions for the authors to consider.

1) The study can be strengthened by using a natural bacterial strain that has a low binding affinity pthA4 TALE, like RVD11 in Table 1. The study could also explore the possibility of PthA4 wild type TALEs to gain virulence on genetically modified LOB EBE resistant plants, which could provide support the authors’ argument about the potential durability of genetically modified plants.

2) The experiment was conducted solely in greenhouse and laboratory. The authors should discuss if the process described in the study could occur in nature. How are the authors going to test?.

3) The authors stated that dTALEs carrying seven mismatches did not have any sequence change after 30 infection cycles. If TALEs tend to have high rate of mutation because of the repeats, why the bacterial strains carrying seven mismatch TALEs did not mutate at all, but those carrying less number mismatches did.

4) Introduction should include some published information about host adaptions by other pathogens, like what other mechanisms underline host adaptions.

5) Fig 1D labeling is not very clear, Fig 3C has so many colors, which is not easy to match with labels. Fig 4A, it would be good to show the DNA sequence for each mutation. Table 3 last two columns need more explanations. Fig S2 needs to explain why there were different sizes of bands in each well. It looks like some had more bands than others.

Reviewer #2: This manuscript reports on an interesting study of adaptive evolution of artificial genes encoding transcription activator-like effectors (TALEs) in the bacterial citrus canker pathogen Xanthomonas citri subsp. citri (Xcc). Starting with a mutant in the tale gene pthA4, which is unable to induce the expression of the LOB1 susceptibility gene, the authors followed the emergence of new bacterial clones that re-gained the ability to induce LOB1. This ability was acquired over time from artificial pthA4 gene variants that were not able to bind to the corresponding promoter element of the LOB1 gene, called EBE[PthA4]. All in all, repeated inoculation into and isolation from sweet orange for a duration of 30 cycles allowed obtaining adapted, LOB1-indicing tale genes from constructs that had two-to-five mismatches with the EBE, but not so from constructs that had seven-to-nine mismatches. The observation and characterization of such novel tale gene variants that evolved in about 1000 generations or less is important information for attempts to generate disease resistant crops against TALE-dependent pathogens.

Despite the general interest, this study can only be considered a first proxy because the artificial tale genes were cloned on a small high-copy number plasmid (pBBR derivative) whereas natural tale genes are encoded on the chromosome or on large conjugative low-copy plasmids. Therefore, mechanisms and speed of evolution can be expected to vary dramatically between the experimental setup in this study and the situation in the natural environment. Another criticism concerns the conclusions about affinities of natural and evolved TALE variants and their potential to induce the downstream gene, which are merely based on one computational algorithm (TALE-NT 2.0). Only two variants were characterized experimentally, which is by far not enough to speculate about adaptive evolution and enhanced affinities/activities of the time course of the experiment. At this point, the study needs to be more quantitative and the different TALE variants should be tested experimentally with respect to reporter gene activation. One may even wonder whether or not high affinity is required at all or if a threshold induction level would suffice for symptom formation?

Additional remarks/corrections:

Line 59: Problem with Ref. Kumar Verma et al., 2018.

Line 83: AvrXa7

Line 148: It is not clear where the Tn5 is exactly inserted, and the reference does not help either because 50 random Tn5 mutants in pthA4 were obtained in that study.

Line 213: It is not clear why PthA4 is not mentioned here? Codons from pthA4 should be present in the strain, even if Tn5 has disrupted the gene. In fact, the study would have been more instructive if the authors had chosen a deletion mutant in pthA4 – or even better a gene replacement with the artificial construct containing codons normally occurring in pth genes – instead of a transposon mutant. The question whether intra- or intergenic conversion leads to new tale variants is indeed very interesting and could have been much better addressed with such an experimental setup.

Line 222: That’s the place where one wonders about real affinities (GUS assays, see above).

Line 231: This is again speculation based on predictions. Why not performing transcriptome profiling using RNA-seq, or at least study these candidate genes by qRT-PCR?

Line 252: It would have been instructive to see a scheme where one can easily follow the mutational events that might have led to the new phtA4 variants.

Line 266: suggested

Reviewer #3: see attachment

**Have all data underlying the figures and results presented in the manuscript been provided?**

Reviewer #1: Yes

Reviewer #2: Yes

Reviewer #3: Yes

PLOS authors have the option to publish the peer review history of their article (what does this mean?). If published, this will include your full peer review and any attached files.

Reviewer #1: No

Reviewer #2: No

Reviewer #3: No

---

## [Decision Letter · Decision Letter 1]

11 Dec 2020

Dear Dr Wang,

We are pleased to inform you that your manuscript entitled "Consequences of adaptation of TAL effectors on host susceptibility to Xanthomonas" has been editorially accepted for publication in PLOS Genetics. Congratulations!

As you will see below, Reviewer #1 has a couple suggestions that you may want to consider as you prepare the final version of the manuscript for the production team (these are optional suggestions, and the editorial team will not need to re-evaluate).

Yours sincerely,

Gregory P. Copenhaver

Editor-in-Chief

PLOS Genetics

Comments from the reviewers (if applicable):

Reviewer's Responses to Questions

**Comments to the Authors:**

Reviewer #1: The authors have addressed my concerns and questions and made corresponding changes in the manuscript. There is only one thing I would like to suggest for current version. The authors added some host adaptation studies in introduction. What is your conclusion about those studies and what question(s) those studies have not addressed? What question(s) this study was going to address? This could be added in the last paragraph in the introduction to indicating the rational of your research work.

Reviewer #3: In this revised version of the manuscript now entitled “Consequences of adaptation of TAL effectors on host susceptibility to Xanthomonas” Teper and Wang have successfully addressed most of the concerns I had raised, either by providing new data or by adding new text which is clarifying few uncertain aspects and giving a better perspective to this study which I find very interesting. In conclusion I am satisfied with the manuscript as it stands in its revised form and I thank the authors for the efforts made in providing such a good work.

**Have all data underlying the figures and results presented in the manuscript been provided?**

Reviewer #1: Yes

Reviewer #3: Yes

PLOS authors have the option to publish the peer review history of their article (what does this mean?). If published, this will include your full peer review and any attached files.

Reviewer #1: No

Reviewer #3: No

**Data Deposition**

http://datadryad.org/submit?journalID=pgenetics&manu=PGENETICS-D-20-01296R1

**Press Queries**

---

## [Editor Report · Acceptance letter]

13 Jan 2021

PGENETICS-D-20-01296R1 

Consequences of adaptation of TAL effectors on host susceptibility to Xanthomonas 

Dear Dr Wang, 

We are pleased to inform you that your manuscript entitled "Consequences of adaptation of TAL effectors on host susceptibility to Xanthomonas" has been formally accepted for publication in PLOS Genetics! Your manuscript is now with our production department and you will be notified of the publication date in due course.

With kind regards,

Melanie Wincott

PLOS Genetics

On behalf of:
